# Preparation and Characterization of DOPO-Functionalized MWCNT and Its High Flame-Retardant Performance in Epoxy Nanocomposites

**DOI:** 10.3390/polym12030613

**Published:** 2020-03-07

**Authors:** Liqiang Gu, Chen Qiu, Jianhui Qiu, Youwei Yao, Eiichi Sakai, Liting Yang

**Affiliations:** 1School of Chemistry and Environment, South China Normal University, Guangzhou 510006, China; gulq123@163.com; 2Department of Machine Intelligence and Systems Engineering, Faculty of Systems Science and Technology, Akita Prefectural University, Yurihonjo 015-0055, Japan; e_sakai@akita-pu.ac.jp; 3Advanced Materials Institute, Graduate School at Shenzhen, Tsinghua University, Shenzhen 518055, China; dafeng_116@163.com (C.Q.); yaoyw72@163.com (Y.Y.)

**Keywords:** epoxy resin, functionalized MWCNTs, DOPO, flame retardant, nanocomposites

## Abstract

In this work, functionalized multi-walled carbon nanotubes (MWCNT) were synthesized by the reaction between acylated MWCNT and 10-(2,5-dihydroxyphenyl)-9,10-dihydro-9-oxa-10-phosphaphenanthrene-10-oxide (ODOPB). The obtained MWCNT-ODOPB was well dispersed into epoxy resins together with aluminum diethylphosphinate (AlPi) to form flame-retardant nanocomposites. The epoxy resin nanocomposite with phosphorus content of 1.00 wt % met UL 94 V-0 rating, exhibited LOI value of 39.5, and had a higher *T*_g_ compared to neat epoxy resin, which indicates its excellent flame retardant performance. These experimental results indicated that MWCNT-ODOPB was a compatible and efficient flame retardant for epoxy resins. Moreover, cone calorimeter analysis showed that the peak heat release rate (pHRR), total heat release (THR) values, and CO_2_ production profiles of the composites decreased with an increase in the additional amount of phosphorus.

## 1. Introduction

Epoxy-based composites have a wide range of applications in various industrial productions such as construction materials, encapsulation resins, metal coatings, electrical components/light emitting diodes (LEDs), high tension electrical insulators, structural adhesives, and more [1,2,3]. However, the poor flame-resistant property of epoxy-based composites severely restricts their applications [4]. Halogen-containing flame retardants are an effective choice to improve the flame retardancy, but they can also cause potential environmental problems by releasing toxic and corrosive gases such as hydrogen halides during combustion [5,6]. Therefore, the development of environmentally-friendly halogen-free flame retardants (HFFRs) has received rapidly growing attention from both academia and industry. Among the phosphorous-based HFFRs, 9,10-dihydro-9-oxa-10-phosphaphenthrene-10-oxide (DOPO)-derived HFFRs are considered to be suitable alternatives because of their excellent flame retardancy [7]. New market demands have continued to promote the development of flame-retardant technology for improving thermal stability, processability, and mechanical properties [8,9]. Novel HFFRs such as DOPO-derived flame retardants are steadily gaining more market share.

It has been proven that the addition of carbon nanotubes (CNT) can obtain better thermal stability, processability, and mechanical properties for plastics [10,11,12,13,14]. Moreover, the ability to graft functional groups onto outside surfaces of carbon nanotubes further diversifies their functions [15,16,17]. For these reasons, functionalized carbon nanotubes with enhanced dispersibility and interfacial interaction in polymers have been applied as group carriers to improve the performance of polymer matrixes [18,19,20].

In this study, DOPO-functionalized multi-walled carbon nanotubes (MWCNT) were synthesized by grafting 10-(2,5-dihydroxyphenyl)-9,10-dihydro-9-oxa-10-phosphaphenanthrene-10-oxide (ODOPB) onto the outside surfaces of acylated MWCNT. The functionalized MWCNT was used as an additive together with nano-sized aluminum diethylphosphinate (AlPi) to form fire-resistant epoxy resin composites. The ER/AlPi/MWCNT-ODOPB nanocomposites with 1 wt % phosphorus content exhibited the best UL 94 rating (V-0) and an upgraded limiting oxygen index (LOI) value of 39.5. At the same time, the as-prepared composites also showed higher *T*_g_ values than neat epoxy resin for a good interaction between epoxy chains and MWCNT-ODOPB. Besides, cone calorimeter tests (CCT) were used to study the fire hazards for MWCNT-ODOPB-modified epoxy resins. The results indicated a clear decrease in a peak heat release rate (pHHR) and a total release rate (TRR) with an increase in the phosphorus content. Furthermore, the CO_2_ and CO production during related tests also exhibited the same trend.

## 2. Experimental Section

### 2.1. Materials

MWCNTs (VGCF^®^-X, average diameter was 10~13 nm) were purchased from Showa Denko Company, Japan. Sulfuric acid (H_2_SO_4_ 98 wt %), nitric acid (HNO_3_, 67 wt %), 4,4-diaminodiphenyl methane (DDM), N,N-dimethylformamide (DMF), tetrahydrofuran (THF), pyridine, thionyl chloride (SOCl_2_), absolute ethanol (EtOH), and benzoquinone were purchased from the Tokyo Chemical Industry Co., Ltd., Tokyo, Japan. DOPO, from Eutec Trading Co., Ltd., Shanghai, China, was dried at 100 °C for 2 h after being purified by recrystallization from EtOH. Nano-sized aluminum diethylphosphinate (AlPi) (average particle size was about 300 nm) were dried at 80 °C for 12 h in a vacuum oven before use, which were purchased from Clariant Chemicals Ltd., Frankfurt, Germany. 2,2-Bis (4-glycidyloxyphenyl) propane (known as diglycidyl ether of bisphenol A, short for DGEBA or ER, epoxy equivalent = 0.51 mol/100 g) were purchased from Diaisheng Epoxy Co., Ltd., Wuxi, China.

### 2.2. Preparation of MWCNT-ODOPB

In this work, we present a three-step procedure, as shown in Scheme 1 to prepare flame retardant MWCNT-ODOPB.

1. ODOPB was synthesized with DOPO and benzoquinone as the starting materials and tetrahydrofuran (THF) as the solvent [21]. DOPO was suspended in THF and heated to 80 °C in a multi-necked round bottom flask equipped with a stirrer, a reflux condenser, and a thermometer. Then benzoquinone was added and stirred at 80 °C for another 8 h. After the reaction, the ODOPB product was filtered, washed with water, recrystallized from THF, and dried in a vacuum oven at 80 °C for 12 h.

2. MWCNT-COCl as an important intermediate was prepared by the reported acyl chlorination reaction [22,23]. MWCNT was acidified with a mixture of HNO_3_/H_2_SO_4_ (1:3 by volume) at 50 °C for 4 h by ultrasonication. The resulting acid-treated MWCNT (MWCNT-COOH) was diluted in water and filtered, washed up to a neutral pH (pH = 7.0), and dried at 80 °C under vacuum for 12 h. Subsequently, MWCNT-COOH was dispersed under ultrasonication for 2 h and then refluxed together with SOCl_2_ and *N*,*N*-dimethylformamide (DMF) (quantities of MWCNT-COOH, SOCl_2_, and DMF were 200 mg, 50 mL, and 1 mL, respectively) at 80 °C for 24 h. After the acyl chlorination reaction, unreacted SOCl_2_ was removed by distillation. The resulting MWCNT-COCl was dried under vacuum at room temperature.

3. Specific amounts of ODOPB and MWCNT-COCl with one drop of pyridine as catalyst were dispersed in DMF in a three-necked round flask and stirred at 80 °C for 12 h under a nitrogen atmosphere. The target product MWCNT-ODOPB was filtered, washed with DMF, and dried under vacuum to a constant weight.

### 2.3. Preparation of ER/AlPi/MWCNT-ODOPB Nanocomposites

The ER/AlPi/MWCNT-ODOPB composites with phosphorus contents of 0.078 wt %, 0.75 wt %, 1.00 wt %, and 1.50 wt % (labeled EC-1, EC-2, EC-3 and EC-4, respectively, as shown in Table 1) were cured using 4,4-diaminodiphenyl methane (DDM) as a hardener, as shown in Scheme 2. The curing procedure was presented as follows: the calculated amounts of MWCNT-ODOPB, AlPi, and DGEBA were mixed in a four-necked round flask under stirring by a mechanical overhead stirrer (20 rpm), and accompanied by ultrasonic dispersion (600 W) at 160 °C for 2 h. Then, the calculated amount of DDM (ER/DDM =100/25, *w*/*w*) was added when the above mixture was cooled to 100 °C. After the degasification procedure, the prepared flame-retardant epoxy resin mixture was poured into a customized mold. Subsequently, the flame-retardant system was cured in a vacuum oven at 140 °C for 2 h and then at 180 °C for 2 h. Lastly, the MWCNT-ODOPB modified epoxy nanocomposites were cooled down to room temperature slowly to prevent cracking.

### 2.4. Instrumental Analysis and Measurements

Fourier transform infrared spectroscopy (FT-IR) spectra were recorded by a Nicolet iN10 MX FT-IR spectrometer (Thermo Fisher Scientific, Inc., Waltham, MA, USA) using KBr pellets at the wavelength range of 400~4000 cm^−1^. X-ray photoelectron spectroscopy (XPS) analyses were measured on a PHI Quantum 5000 system photoelectron spectrometer (Perkin-Elmer Corp., Minnesota, USA) equipped with an Al Kα radiation source. Thermal gravimetric analyses (TGA) and differential thermal analyses (DTA) were performed by a Shimadzu DTG-60/60 H instrument at a heating rate of 10 °C·min^−1^ from 30 °C to 600 °C with a nitrogen flow of 50 mL·min^−1^. Transmission electron microscopy (TEM) images were obtained on a Model H-8100 transmission electron microscope (Hitachi Ltd., Tokyo, Japan). Dynamic mechanical analysis (DMA) performed in the RSA-G2 instrument from TA Instruments (New Castle, DE, USA) was used to study the glass transition temperature (*T*_g_) and storage modulus (E′). The Underwriter Laboratories 94 (UL-94) vertical curing tests were performed according to IEC 60695-11-10:1999 to test the ease of the ignition of flame-retardant epoxy nanocomposites using an FZ-5401 burning instrument (Hanyang Electronic Instrument Co., Ltd., Dongguan, China). The morphology of residual chars was investigated by a Hitachi S-4300 (Hitachi Science Systems Ltd., Tokyo, Japan) scanning electron microscopy (SEM) with a 5-kV acceleration voltage. The cone calorimeter experiments for analyzing the flaming behaviour were carried out on an FTT cone calorimeter (Fire Testing Technology (FTT) Ltd., West Sussex, UK), according to the standard ISO 5660-1.

## 3. Results and Discussion

### 3.1. Characterization of MWCNT-ODOPO

#### 3.1.1. FT-IR Analysis

As shown in Figure 1, the peaks of DOPO in the FT-IR spectrum were assigned as follows: 758 cm^−1^, 906 cm^−1^, and 1238 cm^−1^ (P–O–Ph), 1206 cm^−1^ (P–O), 1594 cm^−1^ (P–Ph), and 2436 cm^−1^ (P–H) [24,25]. In contrast, the absorption peak for the P—H stretching vibration was not observed in the spectra of ODOPB, and MWCNT-ODOPB. This indicated the successful reaction between DOPO and 1, 4-benzoquinone. In the spectrum of MWCNT-ODOPB, a new absorption peak appeared at 1330 cm^−1^, which could be attributed to C–O–C stretching [26]. It indicated that the –P–OH groups reacted with –COCl groups on the outer surfaces of MWCNT. Moreover, peaks at 751 cm^−1^, 925 cm^−1^, and 1196 cm^−1^ corresponding to stretching vibrations of P–O–Ph showed chemical shifts compared with DOPO, which also confirmed the formation of MWCNT-ODOPB [27,28].

#### 3.1.2. XPS Analysis

Figure 2 shows the XPS spectra of MWCNT, MWCNT-COCl, and MWCNT-ODOPB. Important evidence of the acylation reaction on the surface of MWCNT and new peaks for Cl2p and O1s of MWCNT-COCl were observed at 200 eV and 533 eV, respectively [29,30,31]. However, the Cl2p peak disappeared and the P2p peak at 135.5 eV appeared in the spectrum of MWCNT-ODOPB. This change confirmed successful grafting of ODOPB onto the outside surfaces of chloro-formylated MWCNT. The high resolution P2p spectra (Figure 2b) also provided the evidence of attaching DOPO-groups onto MWCNT, where phosphorus species in MWCNT-ODOPB corresponded to P=O (133.6 ev), P-C (134.0 eV), and P-O (134.8 eV) with the ratio of 1:2:1 [32,33].

Furthermore, the detailed analyses of C1s spectrum provided more evidence for the three-step reaction. As seen in Figure 2c, the C-Cl bond for MWCNT-COCl appeared at 289.4 eV, which indicated the successful acylation reaction. The disappearance of this peak in the spectra of MWCNT-ODOPB together with the appearance of C-P bond at 286.8 eV indicated the final nucleophilic substitution reaction. The same conclusions could be obtained by the O1s spectrum. As seen from Figure 2d, the appearance of the P=O bond at 535.2 eV and P-O at 531.6 eV of MWCNT-ODOPB also confirmed the reaction between MWCNT-COCl and ODOPB [34]. In summary, the target flame retardant MWCNT-DODPB was successfully prepared by chemically grafting ODOPB onto the surfaces of MWCNT.

#### 3.1.3. Thermal Analysis

Figure 3 shows the TGA curves of pristine MWCNT, intermediate ODOPB, and the end product MWCNT-ODOPB with a heating rate of 10 °C·min^−1^ in N_2_ atmosphere. As seen from the figure, the pristine MWCNT was hard to decompose with residue char of 96.6% even when the temperature reached 600 °C, whereas ODOPB remained only 4.2%. The weight loss of MWCNT-ODOPB was about 30.7% at the end of the test, which leaves 69.3% residue char. The weight losses of both ODOPB and MWCNT-ODOPB increased synchronously in the temperature range of 350~450 °C, the DOPO-containing layer on the surfaces of nanotubes caused the decrease of thermal stability of MWCNT-ODOPB. The relative number of attached DOPO-groups on functionalized MWCNT walls could be calculated by the TGA curves, according to the research conducted by Ma et al. [35]. For the obtained MWCNT-ODOPB, 29.6 wt % of DOPO-containing groups were grafted onto the outside surfaces. This result was used to calculate the phosphorus content and additive amount of each component.

#### 3.1.4. Morphology

Figure 4 shows the TEM images of flame-retardant MWCNT-ODOPB, and pristine MWCNT as a control for the direct evidence of functionalized preparation. As can be seen from the images, there was a clear difference in the wall thickness between single MWCNT and MWCNT-ODOPB. A closer observation showed more details, which confirmed the functionalization reaction of DOPO-containing groups onto MWCNT. The hollow core of MWCNT appeared light-colored images and the surfaces showed a smooth vertical structure with no visible extra phase adhering to them. The TEM images of MWCNT-ODOPB, by contrast, showed thickened walls because the flame-retardant layers covered the surfaces as a shell. These attachments increased the diameter of nanotubes, which indicated that DOPO-containing groups were grafted onto MWCNT as expected.

### 3.2. Flammability and Residual Char Analysis

#### 3.2.1. UL 94 Tests and LOI Measurement

In this section, the flame retardancy of MWCNT-ODOPB (1 wt %) modified epoxy nanocomposites was first evaluated by UL 94 vertical curing tests and LOI measurements. The test results are summarized in Table 1.

For a neat epoxy resin, the LOI value was 25.0. In contrast, the composite with MWCNT-ODOPB content of 1 wt % exhibited LOI value of 31.5, which indicated good flame retardancy. It was in line with the expectation that the synthetic MWCNT-ODOPB could markedly improve the flame-retardant properties of epoxy resin composites. Accordingly, the epoxy nanocomposites in the presence of MWCNT-ODOPB and AlPi showed further improvements in LOI values: 36.5, 39.5, and 41.2 with phosphorus contents of 0.75 wt %, 1.00 wt %, and 1.50 wt %, respectively. These results confirmed that MWCNT-ODOPB and the AlPi system showed an efficient synergistic flame retardant effect when added into epoxy resins. The composites with phosphorus contents of 1 wt % or more met the UL 94 V-0 rating. This could be attributed to the strong charring layer effectively insulating flame, gas, and heat transfer. The compact charring layer presented the direct evidence that MWCNT-ODOPB promote char formation of EP nanocomposites. The above results indicate that these MWCNT/AlPi epoxy nanocomposites were almost nonflammable in air. Thus, it can be concluded that MWCNT-ODOPB is an efficient flame retardant, which can ensure that modified epoxy resin nanocomposites are nonflammable polymers.

#### 3.2.2. Morphology

Physical structures of charring layers play an important role in preventing droplet generation, flame spreading, and heat transfer. To further study the combustion mechanism of MWCNT-ODOPB-modified epoxy resin nanocomposites, the morphology of char residues after UL 94 vertical burning tests was investigated by SEM. As shown in Figure 5, all the residual char of epoxy nanocomposites presented a rugged carbonaceous layer with various pores distributed unevenly in the burned matrices. Furthermore, the high magnification images show more details about the foam structure, which is a typical morphology of intumescent char residues. This specific structure could be due to the gaseous products generated in the combustion process. According to the flame retardant mechanism of the vapor/condensed phase and free radical quenching [35,36], such a process will produce nonflammable gas and effectively quench phosphorus-free radicals, which can efficiently impede combustion.

#### 3.2.3. Fire Hazards

Cone calorimeter test (CCT) was used for studying the fire hazards of MWCNT-ODOPB-modified epoxy nanocomposites. The resulting curves and related parameters of a peak heat release rate (pHRR), total heat release (THR), smoke production rate (SPR), total smoke production (TSP), CO_2_ production (CO_2_P), and CO production (COP) were presented in Figure 6. As seen from Figure 6a, EC-1 showed a high pHRR value of 837.8 kW/m^2^ at 145 s. The HRR curves of EC-2, EC-3, and EC-4 showed a fluctuation of HRR values between 108–180 s, and presented a remarkable decline in pHRR values to 603.3 kW/m^2^, 433.4 kW/m^2^, and 367.3 kW/m^2^, which is compared with EC-1. The delay and decrease in pHRR values provided the evidence that MWCNT-ODOPB had an excellent synergistic flame-retardant effect with AlPi on the epoxy matrix.

As seen in Figure 6b, THR of EC-2, EC-3, and EC-4 decreased to 83.2 MJ/m^2^, 81.9 MJ/m^2^, and 56.5 MJ/m^2^, respectively, which is compared to 114.5 MJ/m^2^ of EC-1. The lowest THR value was observed for EC-4, which was reduced by 50.7% when compared to EC-1.

Figure 6c,d show the SPR and TSP profiles of MWCNT-ODOPB modified epoxy nanocomposites. The values of SPR and TSP increased slightly following the increase amounts of MWCNT-ODOPB and AlPi.

The asphyxiating properties of CO_2_ and CO associated with burning plastics cause serious injuries, deaths, and increasing challenges for fire rescue. Figure 6e,f, respectively, present CO_2_ and CO production profiles of MWCNT-ODOPB-modified epoxy nanocomposites. The addition of MWCNT-ODOPB and AlPi reduced the pCO_2_P, which indicates a significant CO_2_-release inhibition effect. The pCO_2_P value of EC-4 was reduced by nearly 64% when compared with EC-1. In contrast, the values of pCOP changed only slightly with the increase in additional amounts of MWCNT-ODOPB and AlPi. Reduction of CO may be favorable for a fire rescue because of the reduced toxicity of volatile substances in the burning process.

The excellent carbonization effect of MWCNT-ODOPB in epoxy matrix promoted flame-retardant properties of ER/AlPi/MWCNT-ODOPB nanocomposites. The mass loss versus time curves of MWCNT-ODOPB-modified epoxy nanocomposites are presented in Figure 7. The residual masses (MR) of EC-2, EC-3, and EC-4 were 20.4%, 21.5%, and 27.0%, respectively, which is much higher than that of EC-1 (only 11.7%). Moreover, the slope of the MR curves for EC-2, EC-3, and EC-4 was less than that of EC-1. This was mainly because MWCNT-ODOPB and AlPi formed a char structure during the combustion process of the epoxy composites, which could effectively insulate heat transfer and impede the decomposition. In summary, the flame retardancy of epoxy resin can be improved with modification of MWCNT-ODOPB and AlPi due to their synergistically enhanced flame retardant effect.

### 3.3. Thermal Behaviors

#### 3.3.1. TGA

The thermal stability of epoxy-based functional materials establishes their service environment. For instance, *T*_g_ indicates the crosslinking density of epoxy thermosets, as an important thermal parameter, below which epoxy composites can be used. The thermal stability of ER/AlPi/MWCNT-ODOPB nanocomposites and neat epoxy resin was investigated by TGA at a heating rate of 10 °C·min^−1^ in the N_2_ atmosphere. Figure 8 shows the TGA curves of the thermal decomposition process from 30 °C to 600 °C. There was a nearly synchronous weight loss stage in all curves. However, the enlarged view shows different details for the different samples. The onset decomposition temperatures of 5% weight loss (T_5%_) were 352.8, 350.1, and 347.1 °C for the samples with phosphorus contents of 0.75 wt %, 1.00 wt %, and 1.50 wt %, respectively, which is significantly lower than neat epoxy resin (377.79 °C). This decreasing trend of T_5%_ for DOPO-modified epoxy resin composites could be due to the higher activity of O=P-O bond compared with the C-C bond.

#### 3.3.2. DTA

Figure 9 shows the DTA curves of neat epoxy resin and ER/AlPi/MWCNT-ODOPB composites, and the *T*_g_ values obtained from this figure are summarized in Table 1. Compared with a neat epoxy resin matrix, the composite containing 1 wt % MWCNT-ODOPB exhibited a higher *T*_g_ value of 158.8 °C. This improvement could be due to the addition of functionalized MWCNT, which influenced the molecular chain configuration of the epoxy resin matrix. Moreover, the *T*_g_ values of composites containing AlPi showed a slight increase from 158.2 °C to 174.3 °C with the increase in phosphorus amount. This trend provided evidence of a good synergistic effect between MWCNT-ODOPB and AlPi in an epoxy matrix.

### 3.4. DMA Properties

Figure 10 shows the dynamic mechanical thermal analysis of ER/AlPi/MWCNT-ODOPB nanocomposites, and the results are summarized in Table 1. It can be observed that the addition of AlPi and MWCNT-ODOPB into epoxy resin decreased the storage modulus of epoxy resin (1.94 × 10^9^ Ma). With the increase of phosphorus content from 0.75% to 1.00%, respectively, a storage modulus of modified epoxy resin showed a slight increase for EC-2 (5.86 × 10^8^ Ma) and EP-3 (6.81 × 10^8^ Ma). When phosphorus content came to 1.50%, storage modulus of EC-4 significantly increased to 9.82 × 108 Ma. The *T*_g_ values obtained from DMA showed good agreement with those obtained from DTA.

## 4. Conclusions

Herein, a nanoscale flame retardant MWCNT-ODOPB was successfully prepared by a three-step procedure. The results of FT-IR, XPS, TGA, and TEM analyses confirmed that ODOPB was grafted on the outside surfaces of MWCNT. The as-prepared MWCNT-ODOPB was introduced into epoxy resin together with AlPi to improve the nonflammability of epoxy thermosets. The nanocomposites containing 1.00 wt % MWCNT-ODOPB with the phosphorus content of 1.00 wt % met the UL 94 V-0 rating and exhibited LOI value of 39.5 and high *T*_g_, which indicates its excellent flame-retardant performance. These improvements were attributed to the efficient char residue-promoting ability of MWCNT-ODOPB and synergistic effect between MWCNT-ODOPB and AlPi in an epoxy matrix. Cone calorimeter analysis showed that the pHRR, THR values, and CO_2_ production all decreased with the increase in content of phosphorus.

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
