# Peer review of "Preparation and Characterization of DOPO-Functionalized MWCNT and Its High Flame-Retardant Performance in Epoxy Nanocomposites"

_polymers, 2020, doi:10.3390/polym12030613_

Round 1

Reviewer 1 Report

The author in the manuscript describes the preparation and the characterization of a microencapsulated carbon nanotubes. 

1) The TEM image of MWCNT-ODOPB shows two different thickened walls, what is the plausible reason for such deviation, if the multiwall is uniformily functionalized?

2) Does the author has any control SEM image of the resin?

3)  Thermal decomposition donot show uniform increase in the Tg value. for example, E3 show decrease in Tg, what is the plausible reason? Same thing is observed for DMA.

4) Is there any improvement in the storage modulus with the increase in the phosphorus content. 

Author Response

Reviewer 1

1) The TEM image of MWCNT-ODOPB shows two different thickened walls, what is the plausible reason for such deviation, if the multiwall is uniformily functionalized?

Response 1:

Thanks for the suggestion. It has been proven that the addition of flame retardant functionalized MWCNTs is an effective way to impart good flame retardancy and mechanical properties to nanocomposites.

To ensure the multiwall uniformily functionalized, we focus on two strategies of diameter uniformity of MWCNTs and violent ultrasound chemical reaction conditions for preparation of MWCNT-ODOPB. Firstly, MWCNTs (VGCF®-X) purchased from Showa Denko Company, which average diameter was approximately 10~13 nm. Moreover, as shown in Scheme 1, core-shell nanostructure MWCNT-ODOPB with MWCNTs as hard core and ODOPB as shell was prepared by the following three-step process. Especially, the second and third steps of the preparation reaction are performed under violent ultrasonic conditions.

The TEM images of MWCNT-ODOPB (figure b), by contrast, showed thickened walls as solid tubes because the flame retardant layers covered the surfaces as shell. The different thickened wall shown in the red box was a solid tube, obviously, functionalized with DOPO-containing groups. The reason for such different thickened walls is the different distance to receiver, where the tube in the red box is in bottom layer contrast to the other third ones. Under the above two strategies, We ensure the multiwall uniformily functionalized with ODOPB.

2) Does the author has any control SEM image of the resin?

Response 2:

Thanks for the suggestion. Physical structures of charring layers play an important role in preventing droplet generation, flame spreading and heat transfer. In this study, residual chars was investigated by SEM. While, there were almost none residual chars for flammable epoxy resin, we didnot have the control SEM image of epoxy resin for investigating morphology of residual chars.

Figure S1. SEM images of fractographs from the impact test specimen of epoxy resin

Figure S2. SEM images of fractographs from the impact test specimen of EC-4

However, we can give the control SEM image of the resin for investigating the dipersibility of nano-flame retardant in epoxy resin.

Figure S1 and Figure S2 shows SEM images of the fracture surfaces from impact specimens of epoxy resin and MWCNT-ODOPB modified epoxy resin. It is obvious that there are no clearly bulk aggregates of MWCNT-ODOPB in nanocomposites, which shows well dipersibility of MWCNT-ODOPB in epoxy resin because of the long-time ultrasonication technique.

3)  Thermal decomposition donot show uniform increase in the Tg value. for example, E3 show decrease in Tg, what is the plausible reason? Same thing is observed for DMA.

Response 3:

Thanks for the suggestion. As shown in our previous work[1], associate with adding aluminum diethylphosphinate (AlPi), the glass transition temperature (Tg) showed that Tgs were about 5 °C higher (investigated by DTA), while almost 3 °C lower (investigated by DMA) than that of neat epoxy resin. In this work, the obtained MWCNT-ODOPB was well dispersed into epoxy resins together with AlPi to form flame-retardant nanocomposites. Tgs of these flame-retardant nanocomposites affected by the amount of AlPi and MWCNT-ODOPB added. In the case of the curing system under the game between AlPi (reduce Tg ) and MWCNT-ODOPB (increase Tg), EC-3 showed less sufficiently crosslink against the steric effect during the curing process. Associated with crosslinking density, the glass transition temperature (Tg) of EC-3 was lower investigated by DTA and DMA.

Table S1. Thermal and flame retardant properties of pure-/flame retardant PLA composites.

ID

 AlPi

(wt.%)a

C1

 (wt.%)b

P

 (wt.%)c

Tg

(oC)d

Tg

(oC)e

Char

Yieldf

LOI

UL-94

grade

 Epoxy resin

0

0

0

148.5

161.2

25.4

25.0

Not rating

EC-1

0

1

0.078

158.8

162.6

27.9

31.5

V-1

EC-2

2.92

1

0.75

160.2

172.7

23.3

36.5

V-1

EC-X[1]

4.20

0

1.00

153.7

158.6

20.7

36.0

V-1

EC-3

3.67

1

1.00

156.3

168.5

24.1

39.5

V-0

EC-4

6.18

1

1.50

174.3

180.2

27.5

41.2

V-0

[1] Gu, L., Qiu, J., Sakai, E. (2017). Thermal stability and fire behavior of aluminum diethylphosphinate-epoxy resin nanocomposites. Journal of Materials Science: Materials in Electronics, 28(1), 18-27.

4) Is there any improvement in the storage modulus with the increase in the phosphorus content.

Response 4:

Thanks for the suggestion. As shown in Fig. 11, it can be observed that the addition of AlPi and MWCNT-ODOPB into epoxy resin decreased the storage modulus of epoxy resin (1.94´109 Ma).With the increase of phosphorus content from 0.75% to 1.00%, respectively, storage modulus of modified epoxy resin showed slightly increase for EC-2 (5.86´108 Ma) and EP-3 (6.81´108 Ma). When phosphorus content came to 1.50%, storage modulus of EC-4 increased to 9.82´108 Ma.

Reviewer 2 Report

The paper is very practical in the content and suitable for publication after the authors revised the manuscript.

After reviewing the content of the entire article, the use of ``functionalized`` will be better than ``microcapsulated``. The IG / ID ratio of CNTs after acidification should be smaller because the graphite structure is destroyed. The authors should provide the discussion about the effect of the process of functionalization. The lack of pure CNT data in Table 1 does not show the effect of functionalization. The authors should provide the data. On line 208~212, these contents should be arranged on the left and not centered.

Author Response

Reviewer 2

The paper is very practical in the content and suitable for publication after the authors revised the manuscript.

1) After reviewing the content of the entire article, the use of ``functionalized`` will be better than ``microcapsulated``.

Response 1:

Thanks for the suggestion. We improved the organization of the title as “Preparation and characterization of DOPO-functionalized MWCNT and its high flame-retardant performance in epoxy nanocomposites” to highlight the strategies of searching an efficiency flame retardant DOPO grafted onto MWCNT.

2) The IG / ID ratio of CNTs after acidification should be smaller because the graphite structure is destroyed.

Response 2:

Thanks for the suggestion. We check our previous work [1], IG / ID ratio of DOPO-functionalized MWCNT was decreased the graphite structure is destroyed. It may be the test machine errors resulting in the abnormal IG / ID ratio of MWCNT-ODOPB. We delete section 3.1.2 raman analysis (figure 2).

[1] Gu, L., Qiu, J., Yao, Y., Sakai, E., & Yang, L. (2018). Functionalized MWCNTs modified flame retardant PLA nanocomposites and cold rolling process for improving mechanical properties. Composites Science and Technology, 161, 39-49.

3) The authors should provide the discussion about the effect of the process of functionalization. The lack of pure CNT data in Table 1 does not show the effect of functionalization. The authors should provide the data.

Response 3:

Thanks for the suggestion. We discussed the effect of the process of functionalization in section 3.1 characterization of MWCNT-ODOPO. Compared with pure CNT, functionalized MWCNT grafted with DOPO-containing groups can be indicated that: FTIR chemical shifts of DOPO confirmed the formation of MWCNT-ODOPB; P2p peak at 135.5eV appeared in theXPS spectrum of MWCNT-ODOPB.

Moreover, it has been proven that the addition of flame retardant functionalized MWCNTs is an effective way to impart good flame retardancy and mechanical properties to nanocomposites. Great numbers of cases [2-3] that functionalized MWCNTs served candidates as flame retardants have been reported in recent years, which pure CNT in nanocomposites data were provide as control group. In our work, we focused on the synergistic flame retardant effect of MWCNT-ODOPB and AlPi [4]. Associating with 1 wt.% MWCNT-ODOPB, flame retardancy and thermal properties of epoxy nanocomposites were greatly improved.

Table S1. Thermal and flame retardant properties of pure-/flame retardant PLA composites.

ID

 AlPi

(wt.%)a

C1

 (wt.%)b

P

 (wt.%)c

Tg

(oC)d

Tg

(oC)e

Char

Yieldf

LOI

UL-94

grade

 Epoxy resin

0

0

0

148.5

161.2

25.4

25.0

Not rating

EC-1

0

1

0.078

158.8

162.6

27.9

31.5

V-1

EC-2

2.92

1

0.75

160.2

172.7

23.3

36.5

V-1

EC-X[4]

4.20

0

1.00

153.7

158.6

20.7

36.0

V-1

EC-3

3.67

1

1.00

156.3

168.5

24.1

39.5

V-0

EC-4

6.18

1

1.50

174.3

180.2

27.5

41.2

V-0

 [2] Yu, T., Jiang, N., & Li, Y. (2014). Functionalized multi-walled carbon nanotube for improving the flame retardancy of ramie/poly (lactic acid) composite. Composites science and technology, 104, 26-33.

[3] Xing, W., Yang, W., Yang, W., Hu, Q., Si, J., Lu, H., ... & Yuen, R. K. (2016). Functionalized carbon nanotubes with phosphorus-and nitrogen-containing agents: effective reinforcer for thermal, mechanical, and flame-retardant properties of polystyrene nanocomposites. ACS applied materials & interfaces, 8(39), 26266-26274.

[4] Gu, L., Qiu, J., & Sakai, E. (2017). Thermal stability and fire behavior of aluminum diethylphosphinate-epoxy resin nanocomposites. Journal of Materials Science: Materials in Electronics, 28(1), 18-27.

4) On line 208~212, these contents should be arranged on the left and not centered.

Response 4:

Thanks for the suggestion. We centere the contents on line 208~213.

Round 2

Reviewer 1 Report

The author has made the necessary changes as suggested. I would like to recommend the manuscript for publication.

Reviewer 2 Report

The manuscript is suitable for publication in the journal.